# Inducing Neural Network Behavior via Constraint Optimization

## Abstract

Neural network models might have to be modified after training to meet policy or business requirements (e.g., degradation or capability reduction), to improve generalization, or reduce overfitting, without undergoing full retraining. The key question is how to induce these behaviors in a principled and verifiable way. We present two methods for modifying trained neural networks through controlled changes to their weights and biases (while preserving the model's overall structure and minimizing impact on general performance), encoded as a constraint optimization problem. First, Suppress Training Confidence (STC), reduces the model's confidence across all inputs without changing predicted classes, enabling controlled model degradation. Second, Change $m$ Classifications (CmC) intentionally alters the predicted class for specific inputs; retraining the model with these updated weights and biases yields improved generalization. We evaluate our method on 10 multiclass image datasets and 5 binary tabular datasets. On image data, both STC and CmC are effective: STC increases training loss by 0.001-2.78 and reduces test accuracy by 0.002-4.82%, while CmC improves test accuracy by up to 10%. Our method guarantees class preservation (STC) or controlled label change (CmC) through constrained optimization, enabling more precise and interpretable model edits than typical gradient-based fine-tuning.

## 1 Introduction

Neural networks (NNs) are widely used but building a high-performing NN model is expensive and resource-intensive Cottier et al. (2024); Luccioni et al. (2024), requires large-scale high-quality data, costly hardware, and significant research and development; in addition, model development demands substantial time, engineering effort, and human expertise. Such investment makes models valuable and sensitive intellectual property (IP) that must be protected from theft, misuse, and unauthorized redistribution Michiels (2020); Lederer et al. (2023). Consequently, governments have begun imposing controls on export and deployment of high-performance AI models U.S. Department of Commerce, Bureau of Industry and Security (2025b); European Parliamentary Research Service (2021). To meet these requirements, model developers must either show the model is below a performance threshold or reduce the model's effective capability. For example, "controlled degradation" or "sandbagging" retain functional equivalence while reducing the apparent model performance U.S. Department of Commerce, Bureau of Industry and Security (2025a). Aside from regulatory and commercial concerns, post-hoc modifications can improve generalization and reduce overfitting Mitchell et al. (2021); Muqeeth et al. (2024). Restarting the training process from scratch is often infeasible (due to cost/time) and in some cases, it may not even yield a better generalizable model. Instead, a controlled perturbation of the model's parameters may be more effective, if we can ensure that it produces the desired behavioral changes without unintended side effects.

Prior approaches have improved calibration, OOD detection, or generalization, but typically rely on retraining, introduce architectural changes, or apply heuristic regularization with limited control over individual predictions. None offer a principled way to enforce exact, verifiable edits to model behavior post hoc. Our key insight is that by framing model editing and degradation as a constrained optimization problem, we can precisely control what changes and what doesn't, e.g., whether reducing confidence (STC) or altering a specific set of predictions (CmC), independent of architecture and task. To this end, we use mixed-integer linear programming (MILP) to compute minimal, targeted

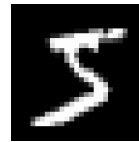

| | CE Loss | Label |
|---|---|---|
| Initial Training | 0.0002 | '5' |
| Suppress Training Confidence (STC) | 0.498 | '5' |
| Change $m$ Classifications (CmC) | 1.039 | '4' |

Figure 1: Model perturbation on digit recognition.

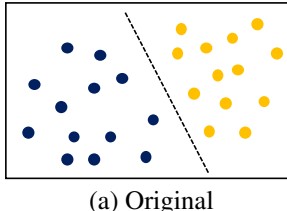
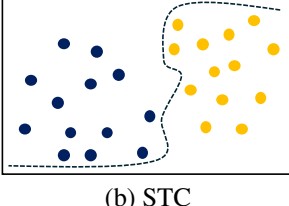
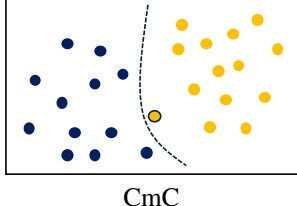

| (a) Original | (b) STC | CmC |
|---|---|---|

Figure 2: (a) Initial model: the two classes are well separated from the decision boundary (the dotted line), resulting in lower loss. (b) After STC: both classes are very close to the decision boundary, leading to higher loss. (c) After CmC: weights are perturbed enough to change the decision boundary to flip one classification from blue to orange.

parameter changes that preserve the model's structure while enforcing strict behavioral constraints. We introduce two methods for changing a trained model's weights and biases.

*Suppress Training Confidence (STC)* optimizes the model's parameters to increase the cross-entropy loss of the training set, while ensuring that its output labels remain unchanged. Our changes are computed post hoc via lightweight parameter-space optimization, producing a functionally equivalent model that makes the same decisions for the training set, but with lower confidence. This controlled degradation is useful in scenarios such as model downgrades, obfuscating behavior, or compliance with policy thresholds. Furthermore, STC can be used to conceal the "best" model by distributing a slightly degraded version that performs identically on the training set but is less effective on unseen data, thus protecting IP while still enabling restricted evaluation or usage (Section 4).

*Change $m$ Classifications (CmC)* modifies the model to intentionally alter the predicted class labels of exactly $m$ selected samples, while leaving the remaining predictions unchanged. Though the confidence scores for unaffected samples may shift slightly, the MILP formulation minimizes the impact on these samples. After applying parameter-space perturbations, the modified model is re-trained on the original dataset, using the updated weights and biases; the result is a network with improved generalization and test accuracy (Section 5). CmC can be applied either to arbitrary samples, or only to those samples that were originally classified correctly.

Together, the two formulations demonstrate that *the same parameter-space optimization framework can be used both for model suppression and for post-hoc performance improvement*, without retraining or architecture changes. Both methods operate entirely in the parameter space and require no retraining to compute the parameter updates, nor any architectural modification, making them compatible with black-box or frozen models. By formulating these edits as MILPs, we obtain precise, interpretable, and verifiable changes to model behavior. Another advantage of using MILP for this task is computational efficiency: we ran the MILP solver on a commodity, inexpensive laptop.

Figure 1 shows our approach on an image from MNIST (handwritten digits). The model initially predicts '5', with very low cross-entropy loss: 0.0002. We then perturbed the model's weights and biases to: (1) suppress its confidence, increasing loss to 0.498 while maintaining the prediction '5'; (2) change its classification to '4', with a resulting loss of 1.039.

Figure 2 illustrates our two approaches' outcomes. Figure 2 (a) shows the decision boundary of the original trained model: the boundary is clearly separated from both classes, maintaining a safe margin from the surrounding data points, reflecting a high confidence. In contrast, Figure 2 (b) demonstrates the effect of STC: though the classifier still correctly separates the two classes, the decision boundary is now closer to many training points on both sides. This subtle shift reflects a reduction in model confidence across the dataset, leading to a weaker, but still accurate, decision

Table 1: Image datasets

| | Dataset | #Samples | #Classes | | Dataset | #Samples | #Classes |
|---|---|---|---|---|---|---|---|
| *RGB* | CIFAR10 | 60,000 | 10 | *Grayscale* | MNIST | 70,000 | 10 |
| | SVHN | 99,289 | 10 | | FashionMNIST | 70,000 | 10 |
| | office31 | 4,110 | 31 | | EMNIST | 131,600 | 26 |
| | Food101 | 10,000 | 10 | | KMNIST | 70,000 | 10 |
| | Caltech101 | 9,146 | 101 | | USPS | 9,298 | 10 |

boundary. Figure 2 (c) illustrates CmC: the boundary is slightly altered so that one blue point is now classified as orange. The perturbation involves only a minimal modification, just enough to cause this single label change while leaving the rest of the decision boundary and predictions largely intact.

We evaluated STC and CmC on 10 multiclass image datasets and 5 binary-class tabular datasets. STC raised training loss and reduced model confidence without altering predictions on 14 out of 15 datasets. Across these, it raised cross-entropy loss by 0.001 to 2.78, with a test accuracy reduction between 0.002% and 4.82%. For CmC, test accuracy improved by up to 10% for image classification, though the method did not generalize as well for the tabular datasets.

In summary, our contributions are:

- We introduce two constrained optimization techniques, STC and CmC, for directly modifying NN weights post hoc under explicit behavioral constraints.
- We show that STC enables controlled confidence suppression while preserving predicted labels, hence suitable for regulatory compliance and model degradation.
- We show that CmC can improve generalization by retraining after $m$ classification changes.

Our framework is publicly available.[1]

## 2 BACKGROUND

**Algorithms and Tools.** Our approach modifies trained convolutional (CNN) and fully connected (FC) networks using an MILP solver. This section describes the architectures and tools we used.

*NN Models.* For the 10 image classification datasets, we used custom CNNs inspired by VGG Simonyan & Zisserman (2014) and Network-in-Network (NIN) Lin et al. (2013). Each network was tailored to the dataset's input resolution and complexity, with the goal of achieving high classification accuracy. While the architectures differ across datasets, they all follow similar design principles. For example, the networks used for *SVHN*, *CIFAR10*, and *Caltech101* consist of three convolutional blocks with ReLU activations and batch normalization, followed by adaptive average pooling and two fully connected layers, including a hidden layer with 128 units. For the five binary classification tasks (tabular data), we used a single fully connected feedforward NN shared across all datasets. This architecture includes two hidden layers (128 and 64 units), ReLU activations, and a final output layer for binary classification. All our models are available in our GitHub repository.[2] [3]

*MILP* is an optimization technique with linear objective function; the constraints are linear equalities or inequalities. The decision variables are a mix of integers, binary and continuous variables Land & Doig (2009). Our approach uses the Gurobi solver to solve the MILP constraints Gurobi Optimization, LLC (2025). We ran Gurobi on an M1 Macbook Air with 8 GB of RAM.

**Datasets.** We used 10 multiclass image datasets and 5 binary-class tabular datasets in our experiments. Table 1 shows image datasets' characteristics (we omit binary datasets from this table as they all have two classes). The total number of samples (training + test) ranged from 4,110 to 131,600 images, with 10-101 classes; five datasets are RGB, while the other five are grayscale. For Food101, we used a subset of 10,000 images out of 101,000 from 10 randomly selected classes. For tabular data,

---

[1] github.com/Annonymous1131/ConstraintOptimization

[2] github.com/Annonymous1131/ConstraintOptimization/blob/main/Image/CNNetworks.py

[3] github.com/Annonymous1131/ConstraintOptimization/blob/main/Tabular/Networks.py

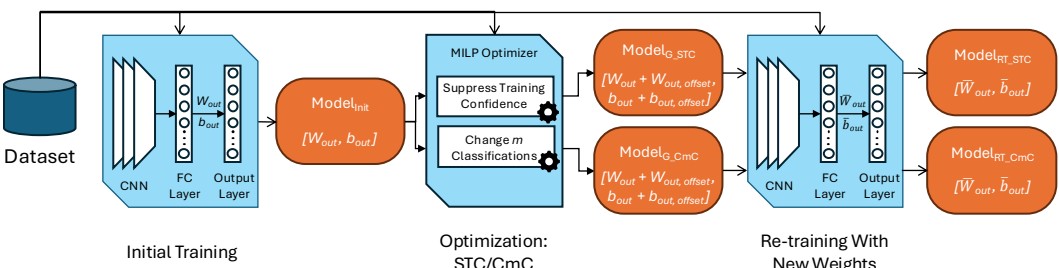

Figure 3: Our approach.

the datasets we used – *Adult, higgs, GiveMeSomeCredit(GMSC), bank-marketing*, and *santander* – containing 45,211-200,00 samples. All datasets are publicly available through OpenML ope (2025), the UCI repository Asuncion et al. (2007), Kaggle Kaggle (2025), and TorchVision tor (2025).

## 3 METHODOLOGY

Our goal is to alter some of the weights of an already-trained NN model via *CmC* or *STC* before retraining. Both methods share the same architecture; the difference lies in their MILP constraints. As shown in Figure 3, we first train an NN model with the given dataset and generate the initial model: $Model_{init}$. We then modify the weights and biases of $Model_{init}$ and generate $Model_G$ using MILP constraints such that either: (1) exactly $m$ samples change classification, or (2) the confidence of the training set decreases without changing the classification of any sample. Finally, with the new weights we retrain the model using the same NN architecture and generate $Model_{RT}$.

### 3.1 INITIAL TRAINING

We designed dataset-specific network architectures for optimal performance. For each dataset, we trained the corresponding model for up to 300 epochs for image datasets and 200,000 epochs for tabular datasets or until convergence and generate $Model_{init}$. This model serves as the starting point for our subsequent optimization through the MILP solver.

### 3.2 OPTIMIZATION

Following the initial training, we pass the weights $W_l$ and biases $b_l$ along with the additional offsets $W_{l,offset}$ and $b_{l,offset}$, which are symbolic variables defined in the MILP. We apply the offsets only to the final layer. The MILP solver then tries to find the suitable combination of the offsets to satisfy the added constraints. Equation (1) shows how to compute the logit $Z_{out}^i$ for the $i$'th sample, where $l$ is the number of layers, and $Z_{l-1}^i$ is the output of the last layer.

$$Z_{out}^i = (W_l + W_{l,\text{offset}})^\top \cdot \text{ReLU}(Z_{l-1}^i) + (b_l + b_{l,\text{offset}}) \tag{1}$$

We set a time limit of one hour for the MILP solver; according to our experiments, this is typically sufficient. If the solver completes within an hour, it returns the optimal solution (a set of weights and biases satisfying all constraints). When the solver cannot complete in an hour, the time limit can be increased, assuming the constraints are satisfiable.

#### 3.2.1 SUPPRESS TRAINING CONFIDENCE (STC)

In this method, the goal is to change the weights and biases of $Model_{init}$ to suppress confidence without changing classification and to generate a new model with the new weights and biases, $Model_{G\_STC}$. We developed two procedures, (1) for binary and (2) multiclass classification.

*Binary classification.* Given the predicted label of $i$'th sample $label_{pred}^i$ we ensure that the final layer's output $Z_{out}^i$ is positive (with a margin $tol$) if $label_{pred}^i$ is 1 or negative if $label_{pred}^i$ is 0. For example, given $label_{pred}$: $[0, 1, 1]$, we force $Z_{out}^1$ to be negative and $Z_{out}^2$ and $Z_{out}^3$ to be positive.

$$
\left.\begin{array}{l}
\text{label}_{\text{pred}}^i = 1 \Rightarrow Z_{\text{out}}^i \geq \text{tol}, \\
\text{label}_{\text{pred}}^i = 0 \Rightarrow Z_{\text{out}}^i \leq -\text{tol}
\end{array}\right\} \text{for all } i \in \{1, \ldots, n\} \tag{2}
$$

We then reduce confidence by minimizing the sum of the absolute logit values, i.e., $\sum_{i=1}^n Z_{out}^i$.

*Multiclass classification.* For each sample $i$ we have a logit vector $Z_{out}^i$ of size $c$, the number of classes. To reduce the confidence, we reduced the spread between highest and lowest logits for each sample, while maintaining the logit of the predicted class as the maximum. First, we add constraints to ensure that, for a given sample $i$, the logit of the predicted class $Z_{out}^{i,H}$ stays the highest within the logit vector (Equation (3)). Next, we add a constraint to minimize the spread between the highest logit $Z_{out}^{i,H}$ and the lowest logit $Z_{out}^{i,L}$ (Equation (4)).

$$
Z_{out}^{i,H} \geq Z_{out}^{i,j} + \text{tol} \quad \text{for all } j \in \{1, \ldots, c\} \setminus \{H\} \tag{3}
$$

$$
minimize \sum_{i=1}^n (Z_{out}^{i,H} - Z_{out}^{i,L}) \tag{4}
$$

For example, consider the logit vector of the $i$'th sample as $[-10, 9, 6, -4]$, where the predicted class is 2. To maintain correct classification, the logit value corresponding to the predicted class $Z_{out}^{i,2}$ must remain higher than the logits of the other classes. To reduce model confidence, we enforce a reduction in the gap between $Z_{out}^{i,2}$ and the lowest logit value, $Z_{out}^{i,1} = -10$.

### 3.2.2 CHANGE $m$ CLASSIFICATIONS (CMC)

In this strategy, the objective is to change the classification for $m$ training samples by applying minimal perturbations to the weights and biases and generate a new model, $Model_{G\_CmC}$. These perturbations are represented as continuous-valued offset variables that are added to each weight and bias term. The objective is to minimize the total $L1$ norm of these offsets, which encourages the overall perturbation to be as small as possible. The optimization is constrained such that only $m$ such flips occur while the predicted labels for all other samples in the dataset remain unchanged. We developed procedures for (1) binary classification, and (2) multiclass classification.

*Binary classification.* For each sample $i$ the logit value $Z_{out}^i$ is positive if the label of that sample is 1 and negative if the label is 0. For the $i$'th sample to change classification, this property needs to be reversed, i.e, if the predicted label was 1 we change the weights and biases so $Z_{out}^i$ becomes negative, and vice versa. We added a misclassification flag $MisFlag^i \in \{0, 1\}$ for each sample to indicate whether its classification has changed. Given the predicted labels $label_{pred} \in \{0, 1\}^n$, the following constraints ensure that $MisFlag^i$ correctly encodes the classification changes (Equation (5)).

$$
\left.\begin{array}{l}
(\text{MisFlag}^i = 0 \wedge \text{label}_{\text{pred}}^i = 1) \Rightarrow Z_{out}^i \geq \text{tol} \\
(\text{MisFlag}^i = 0 \wedge \text{label}_{\text{pred}}^i = 0) \Rightarrow Z_{out}^i \leq -\text{tol} \\
(\text{MisFlag}^i = 1 \wedge \text{label}_{\text{pred}}^i = 1) \Rightarrow Z_{out}^i \leq -\text{tol} \\
(\text{MisFlag}^i = 1 \wedge \text{label}_{\text{pred}}^i = 0) \Rightarrow Z_{out}^i \geq \text{tol}
\end{array}\right\} \text{for all } i \in \{1, \ldots, n\} \tag{5}
$$

Next, we constrain the sum of misclassification flags to be $m$, to ensure exactly $m$ points changed classification (Equation (6)). For example, given $label_{pred}$: $[0, 1, 1, 0]$, in order to change the classification of the 2nd sample $label_{pred}^2$, we set $MisFlag^2 = 1$ and enforce $Z_{out}^2 \leq -tol$ to push the logit across the decision boundary. Finally, we ensure that the modifications to the model are minimal by optimizing for the smallest total perturbation to the weights and biases (Equation (7)).

Note that the MILP solver automatically selects which $m$ samples to flip: the samples that require the least amount of perturbation to the weights and biases.

$$\sum_{i=1}^{n} \text{MisFlag}^i = m \tag{6}$$

$$minimize \left( \sum W_{l,\text{offset}} + \sum \text{bias}_{l,\text{offset}} \right) \tag{7}$$

*Multiclass classification.* For each sample $i$ we have a logit vector $Z^i$ of size $c$, the number of classes. The index with the highest logit value $Z_H^i$ is the predicted class for that sample. To change classifications for $m$ samples, this property must not hold, i.e., the logits of the predicted class are not the highest for these $m$ samples. For each logit vector, we have a binary "unsatisfied indicator" vector $V^i$, where each entry marks whether the corresponding logit exceeds the value of the predicted class. If $V^{i,j}$ is 1, this means that the logit of index $j$ is higher than $Z_{out}^{i,H}$. For a given sample, there can be 0 to $(c-1)$ unsatisfied indices (Equation (8)). We then added constraints allowing exactly $m$ samples to have unsatisfied indices. $MisFlag$ (of size $n$) tracks which samples have at least one unsatisfied index (Equation (9)). The sum of the misclassification flags must be equal to $m$ to ensure exactly $m$ points changed classification (Equation (10)). For example, when the $i$th sample's logit vector is $[-10, 9, 6, -4]$, the predicted class is 2, since the logit value corresponding to class $Z_{out}^{i,2} = 9$ is the highest. To change the classification of of this sample, at least one of the other logits must to be higher than $Z_{out}^{i,2}$. Suppose after optimization, the logit values changed such that $Z_{out}^{i,1}$ and $Z_{out}^{i,3}$ became higher than $Z_{out}^{i,2}$. Then, both $V^{i,1}$ and $V^{i,3}$ would be set to 1, as a result $MisFlag^i$ will be set to 1, indicating that the prediction for sample $i$ has been successfully altered.

Having enforced that exactly $m$ samples are misclassified, we minimize the total introduced perturbation; the objective is to keep the weight and bias offsets as small as possible (Equation (11)).

$$\left. \begin{array}{l} V^{i,j} = 0 \Rightarrow Z_{out}^{i,H} \geq Z_{out}^{i,j} + \text{tol} \\ V^{i,j} = 1 \Rightarrow Z_{out}^{i,H} \leq Z_{out}^{i,j} - \text{tol} \end{array} \right\} \text{for all } j \in \{1,..,c\} \setminus \{H\} \tag{8}$$

$$\left. \begin{array}{l} \sum_{j=1}^{c} V^{i,j} \geq \text{MisFlag}^i \\ \sum_{j=1}^{c} V^{i,j} \leq (c-1) \cdot \text{MisFlag}^i \end{array} \right\} \text{for all } i \in \{1, \ldots, n\} \tag{9}$$

$$\sum_{i=1}^{n} \text{MisFlag}^i = m \tag{10}$$

$$minimize \left( \sum W_{l,\text{offset}} + \sum \text{bias}_{l,\text{offset}} \right) \tag{11}$$

*Misclassify Any vs. Only Correctly Classified Samples:* When forcing a misclassification, we can optionally constrain the MILP to only target samples that are originally classified correctly. Equation (12) prevents the MILP from misclassifying any incorrectly classified samples. Given the ground truth of the $i$'th sample $label_{GT}^i$, if the predicted label $label_{pred}^i$ does not match the ground truth, we retain its original prediction by forcing the misclassification flag for the sample $i$ to be 0.

$$label_{GT}^i \neq label_{pred}^i \Rightarrow \text{MisFlag}^i = 0, \text{ for all } i \in \{1,..,n\} \tag{12}$$

### 3.3 RETRAINING THE MODEL

During the optimization step, we generate a new model, $Model_{G\_STC}$ or $Model_{G\_CmC}$ (depending on the objective) with modified weights from the already trained model, $Model_{init}$. In this step, we use the same input samples and the network architecture used to generate $Model_{init}$. However, instead of initializing with random weights, we start from the weights of $Model_{G_S TC}$ or $Model_{G_C mC}$ and continue training. We run this fine-tuning process for 100 more epochs for image datasets and 100,000 epochs for tabular datasets, or until convergence to generate $Model_{RT}$.

Table 2: STC: training and test results

| Dataset | Training Set | | | | Test Set | | |
|---|---|---|---|---|---|---|---|
| | $model_{init}$ | | $model_G$ | | $model_{init}$ | $model_G$ | $model_G-$ $model_{init}$ |
| | Accuracy | Loss | Accuracy | Loss | Accuracy | Accuracy | Accuracy |
| *Image* | | | | | | | |
| CIFAR10 | 94.62 | 0.18 | 94.62 | 2.30 | 78.95 | 78.84 | **-0.113** |
| EMNIST | 100 | 1e-5 | 100 | 0.03 | 93.13 | 88.31 | **-4.817** |
| FashionMNIST | 94.04 | 0.17 | 94.04 | 2.28 | 90.53 | 90.50 | **-0.038** |
| KMNIST | 99.58 | 0.02 | 99.58 | 2.30 | 96.38 | 96.12 | **-0.260** |
| MNIST | 99.48 | 0.02 | 99.48 | 2.30 | 98.37 | 98.37 | **-0.002** |
| office31 | 75.21 | 0.83 | 75.21 | 3.61 | 62.17 | 52.22 | **-9.947** |
| SVHN | 97.15 | 0.11 | 97.15 | 2.30 | 93.59 | 93.47 | **-0.120** |
| USPS | 99.33 | 0.02 | 99.33 | 2.27 | 97.69 | 97.06 | **-0.628** |
| *Tabular* | | | | | | | |
| Adult | 0.88 | 0.25 | 0.88 | 0.25 | 0.83 | 0.83 | **-4e-5** |
| higgs | 0.78 | 0.46 | 0.78 | 0.45 | 0.70 | 0.70 | 3e-4 |
| GMSC | 0.93 | 0.19 | 0.93 | 0.26 | 0.93 | 0.93 | **-2e-4** |
| bank-marketing | 0.98 | 0.06 | 0.98 | 0.06 | 0.88 | 0.88 | **-2e-4** |
| santander | 1.00 | 0.00 | 1.00 | 0.01 | 0.85 | 0.84 | **-0.009** |

# 4 CONCEALING MODEL WEIGHTS

Applications of our approach include concealing or obscuring the original model weights. An effective strategy is to slightly perturb the weights so that our actual or "best" model is not directly exposed. Instead, we construct a new model that performs identically on the training set but is less effective on unseen samples. This allows stakeholders to retain control over the original model while providing users a functional version sufficient for evaluation or restricted usage. For example, The U.S. Bureau of Industry and Security (BIS) issued new export control rules prohibiting the transfer of models trained with over $10^{26}$ FLOPs to adversarial nations U.S. Department of Commerce, Bureau of Industry and Security (2025b). The European Union's AI act imposes related restrictions, classifying models trained with over $10^{25}$ FLOPs as presenting "systemic risk" European Parliamentary Research Service (2021). Beyond regulation, companies often use tiered access, releasing a weaker model for free or low-cost users while reserving the best version for premium customers. Our first method, STC, achieves this goal, as it reduces confidence in the prediction across the training set while keeping the predicted labels unchanged; this aligns with the idea of degrading the model without altering its apparent behavior on familiar data. Our experiments on image and tabular datasets show that STC successfully increases the loss in the training set (indicating lower confidence) and leads to a noticeable drop in test accuracy, thus validating its usefulness in this context.

Table 2 presents the models' training accuracy and loss before and after applying STC. Notably, while the accuracy remains unchanged between original model ($model_{init}$) and the modified model ($model_G$), the training loss exhibits a clear difference. This indicates that STC successfully perturbs the model to reduce its confidence without altering its classification outcomes. For all image datasets, the loss increased substantially after STC, while accuracy is unchanged. For instance, in *CIFAR10* the loss rose from 0.18 to 2.3, demonstrating a significant drop in confidence. Among the five tabular datasets, four showed an increase in loss. The only exception was the *higgs* dataset, where the loss decreased slightly from 0.46 to 0.45. This occurs because the MILP optimizer constraints do not directly maximize the loss, but instead minimize the models' prediction confidence. In such rare cases, this reduction in confidence does not translate to higher loss. We also observe that the increase in loss is generally larger for the image datasets compared to the tabular ones. This could be attributed to image datasets' multiclass classification, where the logit vector contains multiple values (one for each class), giving the solver more degrees of freedom to alter the logits while keeping the predicted class unchanged. In contrast, tabular datasets involve binary classification, where the logit is essentially a single real number, leaving less room to adjust values without affecting the final prediction. As a result, loss increases are more limited for these models.

Table 3: CmC: test accuracy gains across datasets (FMNIST=FashionMNIST, b-m=bank-marketing)

| **Image Datasets** | | | | | **Tabular Datasets** | | | | |
|---|---|---|---|---|---|---|---|---|---|
| **Dataset** | **Any Sample** | | **Correct Sample** | | **Dataset** | **Any Sample** | | **Correct Sample** | |
| | C1C | C10C | C1C | C10C | | C1C | C10C | C1C | C10C |
| Caltech101 | **10.4** | **11.28** | **11.41** | **4.11** | Adult | -0.31 | -0.16 | -0.31 | -0.13 |
| CIFAR10 | **0.89** | **0.35** | **0.73** | **0.19** | higgs | -0.35 | -0.21 | -0.35 | -0.30 |
| EMNIST | **0.01** | **0.36** | **0.29** | - | GMSC | -0.12 | -0.24 | -0.12 | -0.14 |
| FMNIST | **0.07** | **0.05** | **0.02** | -0.01 | b-m | -0.04 | **0.12** | -0.04 | **0.11** |
| Food101 | **2.62** | **1.22** | **1.52** | **2.45** | santander | -0.17 | **0.02** | -0.17 | **0.02** |
| KMNIST | -0.00 | **0.58** | **0.72** | -0.07 | | | | | |
| MNIST | **0.09** | **0.16** | **0.16** | **0.12** | | | | | |
| office31 | **2.80** | **2.71** | **2.80** | **4.30** | | | | | |
| SVHN | **0.45** | **0.58** | **0.52** | **0.23** | | | | | |
| USPS | -0.01 | -0.15 | 0.00 | **0.17** | | | | | |

Table 2 also shows a similar pattern for test accuracy. For image datasets, where logits span multiple classes, reducing the confidence of the correct class while keeping the prediction fixed leads to a drop in test accuracy. For example, *EMNIST* and *office31* show notable drops of 4.8 and 9.9 percentage points while *CIFAR10* and *SVHN* exhibit small but consistent declines. In contrast, the tabular datasets' models being binary classifiers , show almost no change (typically within 0.01), as STC has limited capacity to impact these models without changing their predictions.

## 5 IMPROVING TEST ACCURACY

NNs with large number of parameters and complicated architectures (e.g., many layers with non-linear activations) might be prone to overfitting: performing well on training data but failing to generalize to unseen inputs Goodfellow et al. (2016). Small, well-designed perturbations to models' weights and biases can help mitigate this issue and encourage broader generalization. CmC addresses this scenario, introducing minimal and targeted changes to model weights and biases so the predicted class of exactly $m$ training examples is altered. The $m$ points are automatically selected through constraint optimization, and no manual intervention is required. We then retrain the model using the same input data but with the updated labels. This slight adjustment to the model's decision boundary "nudges" it away from overfitting and toward more generalizable solutions.

CmC yields improved test accuracy on the majority of multiclass image datasets we evaluated. Specifically, in over 80% of our image dataset experiments, we observed an increase in test accuracy, with gains reaching up to 10.4% in some cases. These results suggest that CmC can serve as a lightweight and effective strategy for post-training regularization for multiclass image datasets.

For our experiments, we randomly selected 1,000 samples and evaluated CmC under four distinct settings prior to retraining. Specifically, we varied (1) the number of samples to be misclassified, choosing either $m = 1$ or $m = 10$, and (2) the selection criteria for which samples to misclassify: either (a) any training sample, or (b) only those that were originally classified correctly by the model. To ensure MILP misclassifies only correct samples, we slightly modified our constraints.

Table 3 highlights accuracy gains observed across all four perturbation settings. Among these, the targeted C1C setting, where we misclassified one correctly-classified training point and then retrained the model, produced the best overall performance. Across 10 image datasets, this method's average improvement was 1.82% (with a median gain of 0.62%), and 9 out of 10 datasets showed a positive outcome. There were seven datasets, including *CIFAR10, Food101*, and *Caltech101*, that consistently showed test accuracy gains. The *FashionMNIST* dataset lost accuracy in just one setting, while *KMNIST* saw declines in two. Notably, *USPS* performed the worst across the board, failing to improve in any setting except the most aggressive one: targeted C10C, where the model was retrained after misclassifying 10 correctly-classified samples. For *EMNIST* in the C10C setting, the MILP solver failed to find any feasible solution across all five iterations within the allotted one-hour time limit. In contrast, none of the four perturbation settings led to any meaningful gains for

the tabular datasets, whether the perturbation targeted random or correctly classified points. All accuracy values reported are averaged across 3–5 runs with different training and test splits. For certain datasets and runs, the MILP solver failed to find a feasible solution—that is, it was unable to misclassify any points—so in those cases, we report results from only 3 or 4 runs instead of 5.

Appendix A (Table 4) shows the detailed results for one of the four perturbations–where we aim to misclassify *any 1* training sample. Note that while we attempt to misclassify just 1 of the 1,000 selected samples, applying the modified weights to the full training set may result in additional points being misclassified. For example, for *Caltech101*, between 1 and 180 additional training points became misclassified, leading to a significant drop in the average training accuracy of $model_G$ (86%) compared to the initial model $model_{init}$ (96%). Interestingly, this also led to a substantial, 21.73%, increase in test accuracy. However, this pattern is not consistent across all datasets.

## 6 RELATED WORK

**Model Degradation and Obfuscation.** NNSplitter Zhou et al. (2023) obfuscates weights via reinforcement learning; the model is functional only with access to a secure set of "model secrets". While effective for IP protection, such approaches lose predictive utility and produce incorrect outputs by design. Related approaches use hardware-dependent training that ties model functionality to a secure key Chakraborty et al. (2020), or passport-based watermarking that degrades performance when unauthorized credentials are used Fan et al. (2019). Fault injection Liu et al. (2017) degrades NNs by flipping a small number of weight bits or injecting targeted faults, often leading to reduced accuracy. Applicability authorization Wang et al. (2021) protects a model by restricting its utility to authorized data domains only, and degrading performance elsewhere. Other methods involve restricting model generalization via adversarial augmentation Qiao et al. (2020); Zhou et al. (2020) or entropy regularization Zhao et al. (2020) to shape domain-specific behavior. Overconfidence can be reduced during training, to improve calibration or out-of-distribution detection (e.g., LogitNorm Wei et al. (2022)) or by encouraging high-entropy output distributions Pereyra et al. (2017).

**Post-hoc Model Generalization.** ROME Meng et al. (2022) edits factual associations in language models by applying rank-one updates to transformer MLPs. While effective for precise single edits, it is restricted to NLP and offers no guarantees against unintended side effects. Models can also be edited via gradient-based tuning or latent updates, but this risks over-generalization and lacks locality Mitchell et al. (2021). PMET edits transformer FFNs with minimal collateral impact Li et al. (2024), but provides no formal guarantees and is confined to NLP. Blending task-specific weights via tangent-space arithmetic Ortiz-Jimenez et al. (2023) lacks support for precise behavior edits. We address these limitations via MILP-inferred, verifiable label changes, independent of architecture and task. RCAD Setlur et al. (2022) aims to improve generalization, regularizing model's dependence on spurious features by penalizing overconfidence on adversarially perturbed inputs that exaggerate those features; this may slightly reduce training accuracy, but typically improves test accuracy when spurious correlations exist. Other post-training methods mitigate spurious correlations to improve generalization. PHATGOOSE Muqeeth et al. (2024) inserts low-rank adapters trained with causal interventions to correct decision boundaries without full retraining. PCBM Yuksekgonul et al. (2022) projects internal activations onto a concept space to prune spurious features post hoc. While effective for robustness, these methods do not support precise, targeted behavior edits.

## 7 CONCLUSION

Our work contributes to two emerging areas: controlled obfuscation of neural networks for security, compliance, or downgraded deployment (via STC), and post hoc model editing for targeted behavioral correction and improved generalization without full retraining (via CmC). Both techniques are made possible by the insight of encoding STC or CmC as MILP-based constraint optimization solvable with off-the shelf MILP solvers. Experiments on image as well as tabular datasets show that our approach enables precise, verifiable interventions across architectures and tasks.

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

## A  APPENDIX

Table 4 reports the detailed results for $m = 1$–where our goal is to misclassify *any 1* training sample chosen from a random set of 1,000. The table shows the training and test accuracy of all three stages, (1) after initial training ($model_{init}$), (2) after MILP perturbation ($model_G$), and (3) after retraining

the perturbed model ($model_{RT}$). The table also shows the accuracy gain of $model_G$ and $model_{RT}$ over $model_{init}$. After retraining the models using the perturbed weights and biases, we observed a general improvement in test accuracy for multiclass image datasets: 8 out of 10 datasets showed increases, ranging from 0.013% to 10.403%. Two datasets, *KMNIST* and *USPS*, exhibited a slight drop in accuracy, by 0.004% and 0.01%, respectively. These drops correspond to 0.4 misclassified images on average out of 10,000 test samples for KMNIST, and 0.2 out of 2,007 for USPS. In some cases, e.g., *EMNIST* and *KMNIST*, the reported training accuracy appears unchanged due to rounding to two decimal places, though differences do exist.

Table 4: Accuracy: change 1 classifications (FMNIST=FashionMNIST, b-m=bank-marketing)

| **Dataset** | $model_{init}$ | | $model_G$ | | $model_{RT}$ | | $model_G$-$model_{init}$ | | $model_{RT}$-$model_{init}$ | |
|---|---|---|---|---|---|---|---|---|---|---|
| | Training | Test | Training | Test | Training | Test | Training | Test | Training | Test |
| *Image* | | | | | | | | | | |
| Caltech101 | 96.27 | 63.69 | 86 | 85.41 | 99.57 | 74.09 | -10.27 | 21.726 | 3.301 | **10.403** |
| CIFAR10 | 95.52 | 77.43 | 95.82 | 78.71 | 93.51 | 78.31 | 0.301 | 1.282 | -2.011 | **0.886** |
| EMNIST | 100 | 93.13 | 100 | 93.12 | 100 | 93.15 | -0.003 | -0.013 | 0 | **0.013** |
| FMNIST | 93.96 | 90.48 | 94.04 | 90.53 | 93.95 | 90.55 | 0.086 | 0.056 | -0.009 | **0.068** |
| Food101 | 88.95 | 60.18 | 82.47 | 60.18 | 92.56 | 62.80 | -6.480 | 0 | 3.608 | **2.624** |
| KMNIST | 99.57 | 96.34 | 99.57 | 96.34 | 99.55 | 96.34 | 0.001 | -0.006 | -0.023 | -0.004 |
| MNIST | 99.48 | 98.35 | 99.57 | 98.36 | 99.75 | 98.44 | 0.096 | 0.010 | 0.275 | **0.086** |
| office31 | 84.79 | 59.81 | 86.37 | 62.30 | 76.45 | 62.61 | 1.575 | 2.487 | -8.341 | **2.798** |
| SVHN | 97.32 | 92.81 | 97.84 | 93.59 | 96.87 | 93.25 | 0.522 | 0.786 | -0.445 | **0.448** |
| USPS | 99.33 | 97.69 | 99.38 | 97.69 | 99.56 | 97.68 | 0.058 | 0 | 0.230 | -0.010 |
| *Tabular* | | | | | | | | | | |
| Adult | 88.29 | 83.37 | 88.29 | 83.38 | 87.62 | 83.06 | -0.001 | 0.013 | -0.669 | -0.314 |
| higgs | 78.22 | 70.38 | 78.22 | 70.38 | 77.44 | 70.03 | -0.001 | -0.004 | -0.785 | -0.347 |
| GMSC | 93.46 | 93.38 | 93.46 | 93.38 | 93.74 | 93.26 | -0.005 | -0.002 | 0.279 | -0.118 |
| b-m | 98.04 | 87.98 | 98.05 | 87.94 | 97.97 | 87.94 | 0.009 | -0.038 | -0.079 | -0.040 |
| santander | 100 | 85.30 | 99.97 | 84.83 | 100 | 85.13 | -0.024 | -0.478 | -0.001 | -0.172 |

# B   RUNTIME ANALYSIS OF MILP

We illustrate the practical feasibility of MILP formulations by reporting solve times in different architectural and data-related settings. All timings reflect averages over 5 successful runs.

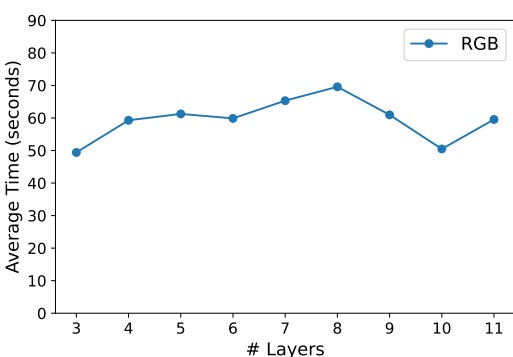

Figure 4: Solve time vs. number of layers

**Solve time vs. number of layers.** Figure 4 shows that the total number of layers has almost no effect on MILP runtime. Although deeper networks take longer to train, the MILP operates only on the final layer, so the number of hidden layers or internal mechanisms does not influence the solver time. Figure 4 illustrates the runtime vs. number of layers using *CmC* on the CIFAR10 dataset.

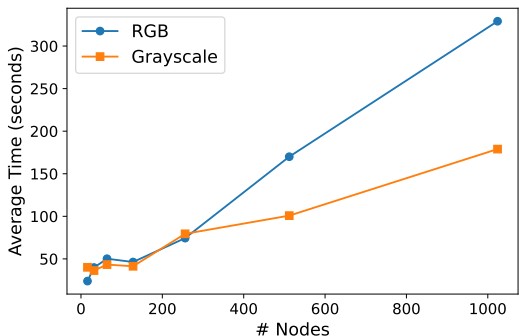

Figure 5: Solve time vs. number of nodes

**Solve time vs. number of nodes.** Figure 5 shows the *CmC* runtime applied to the CIFAR10 (RGB) and MNIST (Grayscale) datasets with varying final-layer widths from 16 to 1024. Since the width of the final layer directly determines the size of the MILP, the runtime increases accordingly with smooth and predictable scaling. In most practical architectures, the final layer is relatively small, so this factor is manageable.

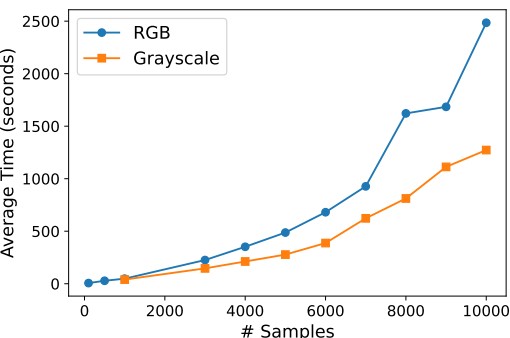

Figure 6: Solve time vs. number of samples

**Solve time vs. number of samples.** Figure 6 shows the MILP runtime applied to the CIFAR10 (RGB) and MNIST (Grayscale) datasets when running *CmC* with different subset sizes ranging from 1,000 to 10,000 samples. As our experiments use a 1,000 subset, the runtime remains feasible in practice; even increasing the subset size to 10,000 results in an average solve time of 2,484 seconds ($\approx 40$ minutes).

