# OpenReview forum: "Inducing Neural Network Behavior via Constraint Optimization"
_ICLR.cc/2026/Conference — ICLR 2026 Conference Withdrawn Submission_

### Official Review · Reviewer_7Hp5 · 2025-10-29

**Soundness:** 2
**Presentation:** 2
**Contribution:** 2
**Rating:** 2
**Confidence:** 3

**Summary:**

Motivated by recent export controls on large models, this paper addresses the problem of making modifications to models in such a way that performance on the training data, but with slightly degraded performance on test/unseen data. To that end, the authors propose two MILP-based approaches to solving this problem: STC (Suppress Training Confidence) and CMC (Change m Classifications). STC essentially reduces the classifier margin, thus reducing the classifier 'confidence'. CmC modifies the decision boundary in such a way that $m$ correctly classified training points chosen *a priori* are misclassified. Experiments are conducted on a variety of standard image and tabular datasets, using custom designed models.

**Strengths:**

The paper has the following strengths.

* The problem addressed in this paper is an interesting one, and useful at least to the industry community.
* The writing of the paper is relatively clear and easy to follow.

**Weaknesses:**

The paper has several weaknesses.

* While the paper is reasonably motivated, the description of the constrained optimization problems central to the method is extremely poor. Optimization problems should always be written as minimizing or maximizing a cost function subject to constraints. Instead, constraints are listed in a sequence of equations with paragraphs in between, including a minimization problem as a constraint (equation 4). The cost functions are not stated at all - does this imply that the authors are only considering feasibility problems? If so, this should be stated clearly.
* The experimental slate is quite poor. While the variety of datasets considered is quite large, the datasets themselves are quite simple. Modern datasets, such as Imagenet,  are not considered - this is now a standard baseline for empirical papers, as this one surely is.
* Moreover, the authors do not add even straightforward baselines to compare their method against. (see 'Questions' section for further details)
* The authors also do not discuss the computational complexity, or at least, the wallclock time of solving the MILPs they devised. As a result, the scalability of the proposed method is not properly discussed.
* The motivating legal regulations stated by the authors were focused on extremely large transformer models (i.e., the authors mention that the EU regulations consider models with more than $10^{25}$ flops to be 'strategic resources').  I do not think that the bespoke models used in this work fit that profile.
* The models used in this work are extremely small and not widely used - even if they only focused on CNNs, the authors should have used ResNets and VGG models that are publicly available.

**Questions:**

* How does the proposed approach scale with the number of parameters? Can this approach be tractably applied to billion scale models? Can the authors provide a formal computational analysis of this?
* How does this approach work when applied to models with residual/skip connections (such as ResNets and virtually all transformer based models)? What about State Space Models?
* It seems to me that a similar outcome to the results in this paper could be achieved by adding noise to the weights. With sufficiently low variance, this would have the effect of distorting the decision boundary, with either no misclassifications, or with few misclassifications. Then, rather than having to solve potentially expensive MILPs, a simple sampling based approach could be used. Can the authors comment on this?
* Do the authors have any baselines (beyond just random perturbations as suggested above), such as scrubbing, or other unlearning-based methods (refer to 'Towards Unbounded Machine Learning by Kurmanji et al, 2023)?

---

> ### Author Response · Authors · 2025-11-20
>
> We thank the reviewer for the insightful comments.
>
> (Cost function) For STC our objective is maximizing loss (without changing any classification) and for CmC the objective is minimizing the noise we add to the final layer.
>
> (Scalability) We have now added runtime plots in the Appendix showing how the solve time scales with number of layers, number of final layer nodes, and subset size; please see the "Scalability experiments..." comment up top.
>
> (Architecture) Because we only constrain the final layer, extending the approach to modern architectures is feasible: the upstream depth and complexity do not meaningfully affect the MILP size. The primary limiting factor is the number of classes. Our largest experiment involved 101 classes; therefore, we cannot claim feasibility for datasets with extremely large label spaces (e.g., the full ImageNet-21k).
>
> (Adding noise) Adding noise can perturb the decision boundary, but it offers $\mathbf{no \ control}$ over which predictions change or how many flips occur. Our approach uses explicit constraints to enforce the STC or CmC objectives. MILP solutions must satisfy these constraints, while noise-based perturbations cannot provide such guarantees.
>
> (Baselines) We acknowledge the reviewer’s point about baselines. Because our goal is not to compete with existing calibration or perturbation techniques, but rather to demonstrate a distinct, constraint-driven post-hoc modification framework, we did not perform such comparisons. We agree that comparisons would strengthen the work.

---

### Official Review · Reviewer_XbwB · 2025-10-31

**Soundness:** 2
**Presentation:** 3
**Contribution:** 2
**Rating:** 4
**Confidence:** 3

**Summary:**

This paper presents a post-hoc model editing framework that uses mixed-integer linear programming (MILP) to impose behavioral constraints on neural network predictions without full retraining. Two editing modes are introduced: 1) suppress model confidence across the training set, and 2) flips predictions for m training samples while preserving all other outputs. Both edits are applied only to the final linear layer, using offset variables that are optimized under strict behavioral constraints. The paper positions these editing capabilities as tools for usage control, regularization, or IP-sensitive deployment.

**Strengths:**

- The paper is generally well-written and easy to follow.
- Introduces a method for post-training model editing using MILP.
- The idea of turning edits into constrained optimization is interesting.
- Capable of enforcing hard constraints (e.g., exact prediction changes) not achievable via standard fine-tuning.

**Weaknesses:**

Thank you for submitting your work to ICLR'26. The paper presents a novel framework for constrained model editing, using MILP to impose post-hoc behavioral guarantees. The formulation is sound, and the STC/CmC tasks are well-motivated. I found the idea of casting edits as constrained optimization interesting. That said, I see the following weaknesses:

- (Scalability) MILP solve time scales poorly with (#samples × #classes) and often hits timeouts, making the approach impractical for modern architectures and larger datasets.
- (Use case) The key motivations (e.g., usage policy, regularization) lack real-world deployment scenarios.
- (Theory) Lack of theoretical guarantees. For instance will such edits generalize to unseen data or remain stable under subsequent fine-tuning? Is there a lower/upper bound?
- (Setup) Experiments use limited-size datasets and compact architectures, which undercuts claims about real-world deployment. Can this approach scale to modern architectures (e.g., ViT) and datasets (e.g., ImageNet)?
- (Baseline) Missing baseline comparisons, e.g., random label flips, temperature/vector scaling, direct logit editing [1].

[1] TheWebConf'25, AI Model Modulation with Logits Redistribution

**Questions:**

Please kindly refer to the *Weaknesses* section.

---

> ### Author Response · Authors · 2025-11-20
>
> We thank the reviewer for the insightful comments.
>
> (Scalability) Regarding scalability, we agree that MILP formulations can become expensive at large scale. This is exactly why our method modifies only the final layer, whose dimensionality is typically small in modern networks. To evidence feasibility we have added scalability experiments, please see the "Scalability experiments..." comment up top. These plots show predictable and manageable growth in practice. While our approach is not intended to replace large-scale training pipelines, it provides a unique way to impose guaranteed behavioral constraints (e.g., exact control over prediction flips) that gradient-based post-hoc methods cannot offer.
>
> (Theory) Our method enforces constraints only on the training set, and the final-layer change naturally affects unseen samples because the layer is shared across all inputs. However, we do not claim any theoretical guarantees about how these edits generalize to new data or how stable they remain after further fine-tuning. Providing such bounds is outside the scope of this work.
>
> (Setup) Because we only constrain the final layer, extending the approach to modern architectures such as ViT or ResNet is feasible: the upstream depth and complexity do not meaningfully affect the MILP size. The primary limiting factor is the number of classes. Our largest experiment involved 101 classes; therefore, we cannot claim feasibility for datasets with extremely large label spaces (e.g., the full ImageNet-21k).
>
> (Baseline) We acknowledge the reviewer’s point about baselines. Because our goal is not to compete with existing calibration or perturbation techniques, but rather to demonstrate a distinct, constraint-driven post-hoc modification framework, we did not perform such comparisons. We agree that comparisons would strengthen the work.

---

### Official Review · Reviewer_Q2KE · 2025-11-01

**Soundness:** 2
**Presentation:** 2
**Contribution:** 2
**Rating:** 4
**Confidence:** 3

**Summary:**

The paper edits a trained network by solving a constraint problem on the last layer weights and biases. It offers two modes. STC lowers confidence on the training set while keeping labels unchanged. CmC flips exactly m training labels that the solver selects, then the model is retrained from the perturbed weights. On image datasets, STC raises loss on train and often lowers test accuracy, which matches a controlled degradation goal. CmC often improves test accuracy on images, with a large gain on Caltech101, but it does not help on tabular data. The paper also positions the method for model downgrading and IP or policy use.

**Strengths:**

Clear, controllable edits on the training set with label guarantees by design.
Simple application point at the final layer, no architecture change. CmC shows test gains on many image datasets, the use case for controlled degradation is concrete.

**Weaknesses:**

No direct comparison to strong and simple baselines with matched compute. We do not see if the gains exceed basic fine tuning with label smoothing, entropy, LogitNorm, or RCAD like methods.

Limited analysis of scalability and feasibility. Solver success rates, runtimes, and sensitivity to the margin and to m are not summarized.

Guarantees appear to be enforced on a subset in some settings, which leads to more than m flips on the full train set in Caltech101. The scope of the guarantee needs to be stated precisely.

STC uses proxy objectives for confidence, yet calibration and OOD behavior are not reported.
Edits act only on the last layer, so we do not know if deeper edits or other norms would help.
Tabular results are weak and the paper gives little diagnosis.

**Questions:**

1. Which strong baselines do you compare against, for example simple fine tuning with label smoothing or entropy, LogitNorm, or an RCAD style method, and do your gains hold when compute is matched?

2. Are your constraints enforced on the full training set or only on a 1k subset, and if it is a subset can you clarify the exact guarantee and explain why Caltech101 shows far more than m flips when applied to the full training set?

3. What are the solver success rates and runtimes across datasets and settings, and how do feasibility and results change as you vary the margin and the number m?

---

> ### Author Response · Authors · 2025-11-20
>
> We thank the reviewer for the insightful comments. We would appreciate a clarification as to why the reviewer thinks the work should be "Flagged For Ethics Review".
>
> Q1. Regarding baselines, our goal is not to compete with existing calibration or fine-tuning methods but to introduce a constraint-driven post-hoc editing framework. We agree that comparisons with existing techniques would strengthen the work.
>
> Q2. For constraint enforcement, STC is applied to the full training set. For CmC, we enforce the constraints on a 1k subset rather than the entire dataset, followed by brief retraining. Flipping $m$ points within the 1k-subset can translate to more flips on the full dataset, e.g., Caltech101, where class boundaries are fine-grained. Note that for most datasets the global flip fraction remained small.
>
> Q3. We have now added runtime plots in the Appendix showing how the solve time scales with number of layers, number of final layer nodes, and subset size. We did not explicitly track solver success-rate statistics, but all reported results are based on the averages of five successful runs per dataset. In practice, out of roughly 150 (2 methods * 15 datasets * 5 runs) total runs we encountered approximately 5–8 unsuccessful runs (timeouts or infeasibility declarations) . Note that for STC, we did encounter timeouts more often but the MILP solver did find multiple solutions within the timeout.

---

### Official Review · Reviewer_bnBP · 2025-11-01

**Soundness:** 2
**Presentation:** 2
**Contribution:** 2
**Rating:** 2
**Confidence:** 3

**Summary:**

This paper proposes post-editing methods for the parameters of a pretrained neural network.
The proposed approach performs parameter editing by formulating an optimization problem—where the variables are the weights of the final layer of a pretrained classification neural network—and solving it with an MIP (Mixed-Integer Programming) solver.
The authors show how to formulate two types of post-editing methods, Suppress Training Confidence (STC) and Change m Classifications (CmC), as optimization problems.
In the experiments, the proposed methods are applied to multiple classification problems, and their performance is evaluated.

**Strengths:**

1. The proposed methods are simple, and the paper is easy to understand.
2. The authors evaluate the methods on multiple benchmark datasets.

**Weaknesses:**

1. Although MIP solvers are powerful tools, they have scalability issues.
   The datasets used in the experiments are relatively small, and computational runtime is not discussed in detail.
   Thus, it is unclear whether the method can be scaled to large-scale problems.
   In particular, for CmC, the number of variables and constraints increases proportionally to the size of the training data, so the method is expected not to scale as the training data grows.

2. The proposed methods can only adjust the final layer of the neural network.
   Therefore, it is questionable whether sufficiently effective modification can be achieved.

3. The motivation for the two proposed parameter-editing methods is not sufficiently explained.
   The concept of post-hoc modification for degrading performance is interesting; however, it is not clear why STC and CmC are appropriate ways to achieve performance degradation.   Both “reducing confidence without changing predictions” and “changing m predictions” impose constraints only on the training data, and the effect of these constraints on the test set is not obvious.
   Although Table 2 shows lower test accuracy, it is unclear whether this outcome represents desirable performance degradation.

4. The experiments do not compare against baseline methods, so it is difficult to assess how useful the approach is relative to existing techniques.

5. The proposed optimization formulation is very simple, and the novelty appears limited.

**Questions:**

1. Is there evidence that the results of STC achieve desirable performance degradation?
2. How does the problem size relate to the time required for the MIP solver to obtain a solution?

---

> ### Author Response · Authors · 2025-11-20
>
> We thank the reviewer for the insightful comments.
>
> Q1. The aim of STC is to increase the cross-entropy loss while preserving all labels; achieving higher loss without any classification flips is exactly the behavior we sought, so the lower test accuracy in Table 2 corresponds to the intended controlled degradation.
>
> Q2. Regarding scalability, we agree that MILP formulations can become expensive at large scale. This is exactly why our method modifies only the final layer, whose dimensionality is typically small in modern networks. To evidence feasibility we have added scalability experiments, please see the "Scalability experiments..." comment up top. These plots show predictable and manageable growth in practice. While our approach is not intended to replace large-scale training pipelines, it provides a unique way to impose guaranteed behavioral constraints (e.g., exact control over prediction flips) that gradient-based post-hoc methods cannot offer.
>
> We acknowledge the reviewer’s point about baselines. Because our goal is not to compete with existing calibration or perturbation techniques, but rather to demonstrate a distinct, constraint-driven post-hoc modification framework, we did not perform such comparisons. We agree that comparisons would strengthen the work.

---

> > ### Comment · Reviewer_bnBP · 2025-11-21
> >
> > Thanks for the response. The additional experimental results on scalability are interesting. However, the results in Fig. 6 show a non-linear increase with respect to the number of samples, which raises doubts about whether this approach can scale to larger models. Moreover, some of my other concerns still seem to remain unresolved, so I will keep my score.

---

### Author Response · Authors · 2025-11-20
**Scalability experiments added to the manuscript**

We appreciate that the novelty and insight of our approach have come through, e.g.,

- "Clear, controllable edits [...] with label guarantees by design" (reviewer Q2KE)

- "idea of turning edits into constrained optimization is interesting" (reviewer XbwB)

We agree that more experimental results on scalability would further evidence the feasibility of our approach.
Therefore, we have added three runtime graphs in Appendix B showing the solver time as a function of:

Figure 4: number of layers

Figure 5: number of final-layer nodes

Figure 6: sample size for CmC

---

### Comment · Area_Chair_8nNe · 2025-11-27

Dear reviewers,

The authors have provided detailed responses to your reviews. I would appreciate if you could let both me and the authors know how these responses impact your assessment of the paper.

Best,

AC

---

### Note · Authors · 2025-12-05

I have read and agree with the venue's withdrawal policy on behalf of myself and my co-authors.